# tDCS and Cognitive Training for Fatigued and Cognitively Impaired People with Multiple Sclerosis: An SCED Study

**DOI:** 10.3390/brainsci15080807

**Published:** 2025-07-28

**Authors:** Teresa L’Abbate, Nefeli K. Dimitriou, George Dimakopoulos, Franca Tecchio, Grigorios Nasios

**Affiliations:** 1Department of Psychology, International Telematic University UniNettuno, 00186 Rome, Italy; teresaabbate95@gmail.com (T.L.); francamatilde.tecchio@cnr.it (F.T.); 2Laboratory of Electrophysiology for Translational Neuroscience (LET’S), Istituto di Scienze e Tecnologie della Cognizione, CNR, 00196 Rome, Italy; 3Department of Speech and Language Therapy, University of Ioannina, 45500 Ioannina, Greece; d.nefeli@uoi.gr; 4Biostats—Statistics & Software, 45110 Ioannina, Greece; info@biostats.gr

**Keywords:** multiple sclerosis, fatigue, cognitive impairment, rehabilitation, non-invasive brain stimulation

## Abstract

**Background/Objectives**: Fatigue and cognitive impairment are common issues for People with Multiple Sclerosis (PwMS), affecting over 80% and 40–65%, respectively. The relationship between these two debilitating conditions is complex, with cognitive deficits exacerbating fatigue and vice versa. This study investigates the effects of a multimodal intervention combining cognitive rehabilitation and neuromodulation to alleviate fatigue and enhance cognitive performance in PwMS. **Methods**: The research employed multiple baselines across the subjects in a Single-Case Experimental Design (mbSCED) with a cohort of three PwMS diagnosed with Relapsing–Remitting MS. The intervention protocol consisted of a baseline phase followed by a four-week treatment involving transcranial direct current stimulation (tDCS) and cognitive training using RehaCom^®^ software (version 6.9.0). Fatigue levels were measured using the modified Fatigue Impact Scale (mFIS), while cognitive performance was evaluated through standardized neuropsychological assessments. **Results**: The multimodal protocol exhibited high feasibility and acceptability, with no dropouts. Individual responsiveness outcomes varied, with two PwMS showing significant decreases in fatigue and improvements in cognitive performance, particularly in the trained domains. Their motor performance and quality of life also improved, suggesting that the treatment had indirect beneficial effects. **Conclusions**: This study provides preliminary evidence for the potential benefits of integrating neuromodulation and cognitive rehabilitation as a personalized therapeutic strategy for managing fatigue and cognitive impairments in MS. Further research is needed to delineate the specific contributions of each intervention component and establish standardized protocols for clinical implementation. The insights gained may lead to more effective, tailored treatment options for PwMS.

## 1. Introduction

Multiple sclerosis (MS) is a complex neurological disease characterized by the immune-mediated demyelination of the central nervous system [1]. This intricate pathology gives rise to a wide range of symptoms, including fatigue and cognitive impairments, which pose a particular challenge for People with MS (PwMS) [2]. Fatigue in MS is a debilitating symptom for more than 80% of PwMS [3], while cognitive impairments in domains including memory, attention, problem-solving, and information processing [2] are estimated to be present in 40–65% of PwMS [4].

The relationship between fatigue and cognition in MS reveals a complex interplay, with the literature showing conflicting results. Whereas several investigations have failed to uncover a significant relationship between self-reported fatigue and objective cognitive test performance, others report that fatigue is linked to cognitive function [5]. The mechanisms underlying the intricate interaction are multifaceted. In particular, fatigue-related processes may directly impact neural networks involved in cognition, leading to cognitive impairments. Conversely, cognitive dysfunction can contribute to increased mental effort and the inefficient use of cognitive resources, resulting in heightened fatigue [6]. The symptom of fatigue has remained rather obscure until more recently, and knowledge about it will improve the ability to uncover its specific interdependence with cognitive impairment. Recent investigations have started revealing that fatigue predicted cognition in PwMS only, beyond anxiety–depressive symptoms and disease progression. These signs highlight the relevance of considering the multidimensional nature of fatigue, possibly revealing fatigue as a predictor of different cognitive processes [7].

Although future advancements will clarify the specificity and the interplay of fatigue and cognitive dysfunctions, it is quite clear that both have a significant detrimental effect on patients’ work status [8], daily activities, and independence [5], as well as their overall quality of life [9,10].

While there have been pharmacological interventions aiming to manage cognitive impairment and fatigue, their efficacy has been inconsistent, with frequent side effects [11,12]. In obtaining comprehensive care for cognitively impaired and fatigued PwMS, non-pharmacological and neuro-behavioral therapies are in development. Cognitive rehabilitation has demonstrated positive effects on cognitive performance [13,14], with observed fatigue relief as well [15].

Considering the structural and functional characteristics of the pathology, the functional damage related to the symptom of fatigue—in the absence of revealable structural alterations of the affected regions—lays the basis for an innovative therapeutic approach with the use of neuromodulation that allows treatments to directly influence the dynamics and excitability of neural networks [16]. A recent review of 49 studies involving 944 PwMS [17] showed that non-invasive brain stimulation is potentially relevant to mitigate fatigue, with a clear advantage of transcranial direct current stimulation (tDCS) over repetitive transcranial magnetic stimulation (rTMS) being identified.

Moreover, the neurophysiological counterpart of neuro-behavioral therapies reveals an adaptation linked to cerebral neuroplasticity mechanisms, whereby increasing intra- and inter-hemispheric functional connectivity stimulates the areas and networks actively involved in the various domains of cognition [18]. Thus, a recent systematic review and meta-analysis [19] revealed that tDCS has a favorable effect on cognitive processing speed and fatigue in MS. However, the effects on cognition and fatigue vary based on the specific assessment used.

This study aims to evaluate the feasibility, acceptability, and individual-level responsiveness of a multimodal intervention combining tDCS and cognitive rehabilitation in PwMS with fatigue and cognitive impairment. Through multiple baselines across subjects in a Single-Case Experimental Design (mbSCED) [20] framework, we sought to achieve the following: (1) assess the protocol’s feasibility and the participants’ acceptance, (2) examine individual responsiveness patterns to inform future personalized intervention strategies, and (3) provide preliminary evidence for the potential benefits of this integrative approach in managing MS-related fatigue and cognitive symptoms. SCEDs enable high-quality research with small numbers of participants and allow an intervention to be tailored to the unique person’s needs and to assess its response through a rigorous methodology. Furthermore, SCEDs are flexible regarding the implementation of the intervention because their underlying goal is most often to determine *“Which intervention is effective for this case?*”, and as a result, departing from the initially planned protocol or intervention is allowed, as long as this is explained. This allows for the creation of a suitable protocol that may be given to a greater number of patients in the future.

## 2. Materials and Methods

### 2.1. Study Design

The current study, as described above, employed the mbSCED, which is an n-of-1 trial; these investigations treat a person as the sole unit of observation in a study in order to pursue and attain the best treatment for each person [21]. SCED, in particular, is a set of experimental methods that can be used to test the responsiveness of an intervention on a small number of people (typically one to three), involving repeated measurements, the sequential introduction of an intervention, and specific data analysis and statistics [20]. The SCED protocol was chosen primarily because it allows for high-quality research with a small number of participants. In particular, SCEDs evaluate the effects of an intervention on people individually, which is an important feature for MS, given that the disease could affect any part of the central nervous system; thus, people’s clinical profiles are often diverse, and each PwMS is unique with distinct characteristics. Moreover, studying fewer subjects but more intensely and comprehensively allows insight to be gained into interventions’ mediating effects and the obtainment of better knowledge of the studied subjects and also detects an intervention effect within the (often large) variability of a subject’s performance (e.g., due to fatigue) [22].

The mbSCED, with replication across the participants, focused on fatigue relief enrichment through a well-standardized cognitive rehabilitation protocol. According to this proposal, at least 3 subjects are needed. All the participants begin the intervention with a baseline phase (i.e., without any intervention), during which the target variable (i.e., fatigue) is measured repeatedly for each patient until one of the participants starts the intervention, while the others continue without intervention, with all the participants still being measured repetitively on the target variable.

Therefore, each PwMS started with a non-intervention baseline phase, during which their fatigue level was measured repeatedly for at least two weeks through the modified Fatigue Impact Scale (mFIS). After the baseline phase, the first enrolled person underwent the basic neuropsychological evaluation at the Neurocare Centre in Ioannina, and the following week the multimodal treatment began. This treatment lasted four weeks; the first week consisted solely of neuromodulation, while the next three focused only on cognitive rehabilitation. The tDCS was held in a bright, quiet room of the Neurocare clinic, whereas the cognitive rehabilitation was performed in the participants’ homes. Their fatigue was monitored weekly for the entire duration of the protocol. The week after the end of the multimodal intervention, as well as three weeks later, participant n.1 underwent a neuropsychological evaluation at the clinic. The same process was repeated for the second PwMS with a one-week delay, resulting in three weeks of baseline, and finally it was repeated for the third the following week, with 4 weeks of baseline (Figure 1).

The intervention sequence was designed based on neuroplasticity principles and clinical guidelines. The tDCS was administered first to modulate cortical excitability and prime neural networks involved in fatigue regulation, particularly the bilateral somatosensory cortices. This neuromodulation was hypothesized to create optimal conditions for subsequent cognitive training by enhancing neural plasticity and reducing baseline fatigue levels that could interfere with learning [23,24].

### 2.2. Participants

GN, a neurologist, identified 15 PwMS previously diagnosed with Relapsing–Remitting MS according to McDonald’s criteria [25,26]; 10 of them reported fatigue and cognitive deficits. From this cohort, 3 were randomly recruited for the project via randomizer.org software [27]. The eligibility criteria included the following: age (between 21 and 60 years old), level of education (of at least 6 years), Expanded Disability Status Scale (EDSS) score (between 0–5), fatigue (measured via mFIS > 35 at least two times with a week interval [28]), cognitive deficit (measured via the Symbol Digit Modality Test (SDMT) < 40 [29]), being a native speaker of the Greek language, and normal hearing and vision normal or correct to normality. The exclusion criteria were as follows: relapse in the last 3 months, cognitive rehabilitation or specific interventions against fatigue in the last 3 months, psychiatric disorders, drug or alcohol abuse, pregnancy (confirmed or presumed), other neurological disorders or clinical history of (e.g., epilepsy), and intracranial metal implants.

The study was conducted following the Declaration of Helsinki ethical principles. The ethics committee of the University of Ioannina approved the study protocol (Reference number 7047/29-02-2024). All the participants were informed of the nature of the study and provided written consent for their participation. The participants were notified that they could drop out of the study at any time as they wished, without any impact on their medical treatment. The profile of the study participants is presented in Table 1.

### 2.3. Procedure

Two PhD students (TL, NKD) and a Bachelor student (KS) trained for the uniform administration of tests, neuromodulation, and RehaCom rehabilitation, and they conducted the protocol in a home-based clinical setting under expert supervision (FT and GN).

The protocol included three assessments. These were planned at pre-intervention, 1-week post-intervention, and at 3-week follow-ups, after which they were involved in the evaluation of motor and cognitive performance via objective measures. In particular, the 9-Hole Peg Test (9HPT) [30] was used to assess fine manual performance. Fatigue, mood, quality of life, and user experience were evaluated with self-reported measures. The cognitive assessment was conducted with the Montreal Cognitive Assessment (MoCA) [31] and the Brief International Cognitive Assessment in MS (BICAMS), which includes the SDMT [32], the Greek Verbal Learning Test-II (GVLT-II) [33], and the Brief Visuospatial Memory Test-Revised (BVMT-R) [34]. Fatigue was monitored via a Google form weekly using the mFIS [35]. Furthermore, depression was evaluated with the Beck Depression Inventory-II (BDI-II) [36], and quality of life was assessed with the Greek MS Quality of Life Questionnaire (MSQoL-54) [37,38]. Finally, user experience was rated by both PwMS and therapists with the User Experience Questionnaire (UEQ) [39], considering classical usability aspects (efficiency, perspicuity, dependability) and user experience aspects (originality, stimulation).

The neuromodulation for fatigue was carried out in the first week after the baseline phase, using a commercialized wireless tDCS cap (Platowork, Platoscience, Copenhagen, Denmark). Based on the literature [40,41], the stimulation protocol consisted of five sessions, held on 5 consecutive days, 15 min a day, with an intensity of 1.5 mA (Figure 2A,B). To facilitate technological aspects, avoiding sharing a prototype device between the Italian and Greek laboratories, we decided to adopt a CE-marked system. We used the Platowork device, which, due to its characteristics, was best suited among commercially available devices for stimulating anodally parietal regions versus the occipital cathode. We specifically settled the positioning of the headset based on previous experience for fatigue mitigation [40,41,42,43]. Cognitive rehabilitation began the following week, lasting three weeks, three days a week, for 45 min a day, utilizing RehaCom^®^ software; this training program has demonstrated a beneficial effect on cognitive functions in PwMS [44]. The cognitive domains stimulated were attention, memory, and executive functions. In each session, PwMS performed 3 different exercises (15 min per exercise) that stimulated specific processes of the mentioned domains (Figure 2C). The order of the exercises was randomized among the participants via randomizer.org. All the participants started from the simplest level of difficulty, which progressively increased, determined by the software based on the participant’s average performance in that specific exercise (see Appendix A).

While the RehaCom software automatically adjusts task difficulty based on user performance, our outcome analysis relied exclusively on standardized neuropsychological assessments administered at pre-intervention and the 1-week and 3-week follow-ups. This methodological choice ensured clinical validity, comparability with the existing literature, and objectivity through researcher-administered, rather than algorithm-generated, metrics. Platform-generated performance data were not included in the final analysis to maintain the focus on the validated outcome measures.

### 2.4. Statistical Analysis

The analysis followed a multidimensional approach to assess the feasibility, efficacy, and acceptability of the intervention and involved the visual inspection of the level, trend, and variability between phases [45]. The data were analyzed using https://manolov.shinyapps.io/Overlap/ software (accessed on 1 May 2024), as well as Microsoft Excel 2016.

*Feasibility:* A qualitative analysis of the feedback received at the end of the protocol was conducted, and dropouts were evaluated.

*Individual responsiveness:* For all the tests, we used the Minimal Clinically Important Difference, defined as the smallest change in scores that identified the PwMs as a responder [46]. The participants were considered responsive to the tDCS intervention if they showed a change of >20% of the baseline level. For the tests, the participants were considered responders if they showed a change of >20% in the 9HPT [47], a change in the range of [2.9–5] points in BICAMS [48], MoCA [49], and BDI-II [50]; and a change of 10 points in the MSQoL-54 [38].

*Acceptability:* For the qualitative analysis, daily during neuromodulation and weekly during cognitive rehabilitation, the participants answered the questions “*Did it bother you?*” and “*Was it difficult to use the tDCS/RehaCom program this week?*” on a Likert Scale of 1 to 10 (1 = not at all, 10 = very much). Furthermore, a User Experience Questionnaire was administered [39].

## 3. Results

### 3.1. Feasibility

All the PwMS completed the multimodal intervention with no dropouts. TL and NKD maintained constant contact with the participants via email or telephone throughout the project, fostering a supportive relationship.

### 3.2. PwMS n.1

#### 3.2.1. Individual Responsiveness

*Fatigue:* The fatigue symptom monitoring showed a worsening at the end of the tDCS intervention (23%) and a mitigation at the end of the RehaCom rehabilitation period (−23%), which disappeared at the 1-week and 3-week follow-ups (Table 2). Looking at the behavior of the fatigue symptom overall, there was a worsening during the four weeks of intervention (Figure 3).

*Neuro-psycho-motor assessment:* Motor performance on the 9HPT improved in the 1-week follow-up for both hands (−20% dominant, −26% non-dominant); however, there was a slight deterioration during the 3-week follow-up in the dominant hand (Figure 6A, Table 3).

The MoCA score increased significantly by 5 points in the 1-week follow-up, with a decline of −3 points in the 3-week follow-up (Figure 6B, Table 3). Regarding the BICAMS subscales, the SDMT showed a non-significant trend of improvement, and the GVLT-II performance remained stable with a slight significant improvement at the 3-week follow-up. The BVMT-R demonstrated a progressive improvement of eight and nine in the 1-week and 3-week follow-ups, respectively (Figure 6C–E, Table 3).

The MSQoL-54 showed a tendency of improvement in the Physical Composite Score (PCS) from the pre-assessment to the 1-week follow-up, which was almost entirely lost at the 3-week follow-up, and a significant improvement in the Mental Composite Score (MCS). Lastly, the BDI-II score improved in the 1-week follow-up (−2) but worsened significantly at the 3-week follow-up (Table 3).

#### 3.2.2. Acceptance

During neuromodulation, PwMS1 did not feel discomfort, while, regarding cognitive rehabilitation, they reported problems with concentration, attention, and physical fatigue (see the Appendix B). All the UEQ values (post-Tdcs, 1-week, and 3-week follow-ups) were positive. In particular, for PwMS1 and therapist 1, the highest mean value is for the “perspicuity” feature (M = 2.08; SD = 1.18; M = 2.08; SD = 0.95) (see the Appendix C).

### 3.3. PwMS n.2

#### 3.3.1. Individual Responsiveness

*Fatigue:* The fatigue symptom showed great improvement; the difference was significant between pre- and post-RehaCom intervention (−77%) and in comparisons between the baseline, the 1-week (73%), and 3-week (68%) follow-ups (Figure 4, Table 2).

*Neuro-psycho-motor assessment:* The motor performance on the 9HPT improved in the 3-week follow-up (−20%) only for non-dominant hands (Figure 6A, Table 3). The MoCA score increased significantly (four points) (Figure 6B, Table 3). Regarding BICAMS, the SDMT showed a significant improvement at the 1-week follow-up (17 points) and at the 3-week follow-up (23 points), and the GVLT-II performance had a slight significant improvement at the 1-week follow-up (3 points) and a significant improvement at the 3-week follow-up (10 points), whereas there was a difference between the two follow-ups (7 points). Finally, the BVMT-R demonstrated a small, significant improvement of three and four points, respectively, in the comparison between pre-rehabilitation and the follow-ups (Figure 6C–E, Table 3).

A significant improvement was noted in the MSQoL-54, with a difference in the PCS from the pre-assessment to the 1-week follow-up (14 points), and in the MCS from the pre-assessment to the 1-week follow-up (35 points) and from the pre-assessment to the 3-week follow-up (20 points). Finally, the BDI-II score improved from the pre-assessment to the 1-week follow-up (−15 points) and the 3-week follow-up (−4 points) (Table 3).

#### 3.3.2. Acceptance

During neuromodulation, PwMS2 did not feel discomfort, while, for cognitive rehabilitation, they reported frustration related to a specific module (see the Appendix B). All the UeQ values were positive. In particular, for PwMS 2, the highest mean value was for “attractiveness” (M = 2; SD = 0.76), and for therapist 2, it was for “perspicuity” (M = 2.59; SD = 0.52) (see the Appendix C).

### 3.4. PwMS n.3

#### 3.4.1. Individual Responsiveness

*Fatigue:* For PwMS3, the fatigue scores in week 4 (baseline phase) and week 9 (intervention phase) were treated as outliers because they were two standard deviations away from the mean score. For week 4 and week 9, the values’ average score from weeks 1 to 3 and from weeks 5 to 8 was calculated, respectively. There were significant differences in the comparisons between the baseline and post-tDCS (−20%), as well as between the baseline and the 3-week follow-up (−29%) (Figure 5, Table 2).

*Neuro-psycho-motor assessment:* Motor performance improved significantly for the non-dominant hand between pre-intervention and the 1-week (−21%) and the 3-week follow-ups (−19%) (Figure 6A, Table 3). Cognitive performance was not enhanced significantly based on the MoCA score, which increased only a little between pre-intervention and the 1-week follow-up (2 points) (Figure 6B, Table 3). Nevertheless, in the BICAMS, the SDMT score was significantly improved by five and nine points between pre-intervention and the 1-week and the 3-week follow-ups, respectively; the GVLT-II performance had an improvement tendency from the pre-assessment to the 1-week follow-up (7 points) and a significant improvement in the comparison between the pre-assessment and the 3-week follow-up (13 points), and the BVMT-R showed a significant improvement between the pre-assessment and the 1-week and the 3-week follow-ups (5 points) (Figure 6C–E, Table 3).

#### 3.4.2. Acceptance

During neuromodulation, PwMS3 did not feel discomfort. During the cognitive rehabilitation phase, he reported health and work problems during the last week; these problems affected his performance. All the UeQ values were positive. In particular, for both PwMS 3 and therapist 3, the highest mean value was for “*perspicuity*” (PwMS3-M = 1.42; SD = 0.76; therapist 3-M = 1.91, SD = 0.63) (see the Appendix C).

## 4. Discussion

Through this mbSCED study, it has been highlighted that the synergistic combination of neuromodulation and cognitive rehabilitation seems to be feasible and well-accepted by the people involved. The overall goal was to pave the way for a person-based understanding of the potential benefits, challenges, and future directions of this integrative approach, ultimately contributing to the evolution of effective strategies by personalizing the therapeutic intervention for people facing MS.

### 4.1. Feasibility

The proposed multimodal and multi-setting therapeutic protocol was deemed feasible. Despite the participants’ heterogeneity in terms of personal, clinical, and social profiles, the absence of dropouts over many months indicates that the intervention was well-tolerated and perceived as being overall beneficial.

A significant aspect of this study was the consistent weekly interaction between the therapists and the participants, which created a strong therapeutic alliance, which has been observed to enhance rehabilitation outcomes [51]. Another point to emphasize is the feature of “flexibility” related to remote and asynchronous rehabilitation [52]. The participants appreciated the opportunity to manage their weekly cognitive rehabilitation plan. Consistently, when asked at the final follow-up about their rehabilitation setting preference, all the participants chose tele-rehabilitation, and two out of three also selected “home setting with a clinician coming home”, while none of them preferred the clinic setting.

### 4.2. Individual Responsiveness

Although there was no significant difference in the fatigue scores before and immediately after tDCS, this study highlighted a longer-term positive effect on fatigue and cognitive and motor performance. The inclusion of cognitive training as a part of the treatment does not allow us to discern the neuromodulation’s role; however, it is known that tDCS in combination with other treatments leads to greater clinical and metabolic changes than cognitive training alone [23,53,54].

The consistency of the time behavior of the three subscales with the total mFIS score underlined the multifactorial nature of the personal experience of the symptom [55,56].

Regarding cognitive performance, PwMS showed a significant and progressive improvement in all the stimulated dimensions. The MoCA values assessing the global cognitive performance improved; however, the gain at the 1-week follow-up was partly lost at the 3-week follow-up. This can be explained by the nature of the test, which investigates multiple cognitive domains, some of which were not trained during the RehaCom protocol [57,58]. In agreement with this hypothesis, the values of the BICAMS subscales investigating the same cognitive domains that were trained had a pattern of progressive and significant improvement, with a smaller further improvement at the 3-week follow-up compared to the 1-week follow-up. The positive findings regarding the enhancement and the long-lasting effects of the treatment on attention, processing speed, and executive function in the present study are in concordance with several other studies that have used RehaCom in cognitively impaired PwMS [59,60]. It is important to note that research has shown that cognitive training has the potential to modify the activity of trained neuronal system areas [61,62].

In addition, the multimodal treatment led to an overall improvement in motor performance. The trained cognitive domains are crucial for motor programming and execution; hence, their improvement helps in the organization and sequencing of motor actions. Moreover, the participants performed computer exercises directly involving visual–motor coordination. The neurophysiological counterpart of this interdependence between motor and cognitive performance is linked to working memory, which makes up a network of neural fibers that allows the exchange of information between the motor and cognitive regions [63]. It can be hypothesized that the tDCS promoted an increase in intra- and inter-hemispheric communication, which is useful to improve motor performance [64]. It is important to underline that the non-dominant hand had the most significant improvement. Typically, baseline performances with the non-dominant hand are always worse than those with the dominant hand; thus, the improvement margin is typically more pronounced [65,66].

Finally, during the follow-up of the participants, a progressive improvement in their quality of life was observed. Specifically, the mental component of the quality of life questionnaire was enhanced in all the participants, while the two younger ones with the lower EDSS scores also ameliorated the physical component. Overall, signs emerged of individual empowerment by improving cognitive and motor performance and, consequently, the sense of agency and independence [19,67] supporting personal resilience [68].

### 4.3. Acceptance

Regarding the tDCS treatment, all the participants reported no side effects or annoyance during or after stimulation—a statement that further supports the safety of the device [69,70] and the protocol used [71]. A part of this acceptance is probably due to the use of a commercial device that is ergonomic and user-friendly.

Concerning the RehaCom protocol, the PwMS reported experiencing mental fatigue and difficulty maintaining concentration during and following the completion of the session, albeit due to differing underlying factors.

The user experience measure indicated that both the PwMS and the therapists (TL, NKD, and KS) mostly liked the pragmatic feature of the “Perspicuity” of the protocol. This feature is quantified by answering the following question: “*Is it easy to get familiar with the product and to learn how to use it?*” [39]. The familiarity and the ease of use of a tool benefit both the therapist and the patient, fostering confident application and enhancing the therapeutic effectiveness.

All the stages of the treatment were accompanied by step-by-step guidance via requests and real-time feedback, and this facilitated the PwMS’s participation and confidence. The fatigue monitoring module, the Platowork app, and the RehaCom software were built with “User-Centrated Design” to make interaction intuitive and accessible for everyone, regardless of their level of “digital education”. Last but not least, the personalization of the protocol responded to the participants’ expressed needs by always proposing achievable goals and thereby increasing their adherence to the treatment [72].

### 4.4. Multicenter Collaboration Strengthened Personalized Strategies

This study was conducted by an international Greek–Italian multicenter team, with a collaboration that proved particularly enriching thanks to a fruitful exchange of complementary expertise. One group adopted the tDCS neuromodulation strategy for fatigue mitigation developed by the other, while the latter integrated the study design, an area of expertise contributed by their colleagues from the partnering country. Sharing a longstanding commitment to treatment personalization, the two teams, through the integration of multimodal approaches and the SCED framework, took a first step toward creating tools for developing personalized interventions, enabling the fine-tuning of treatments already shown to be effective within symptomatically heterogeneous populations.

### 4.5. Study Limitations

The present study had several important limitations that should be acknowledged. First, an element inherent to the SCED strategy is that while it provides robust insights into individual-level responsiveness and supports the development of personalized interventions in similar clinical conditions, it does not offer statistical generalizability to broader populations. The small sample size, while appropriate for the SCED methodology, limits our ability to identify subgroups of responders or establish population-level effect sizes that would inform large-scale clinical implementation.

Second, a significant limitation of our study design is the inability to determine the independent contributions of the tDCS versus cognitive training or to assess their optimal sequencing. The chosen sequence (tDCS followed by cognitive training) prevented us from establishing whether observed improvements resulted from tDCS alone, cognitive training alone, or their synergistic interaction. Future studies should employ randomized controlled designs comparing different intervention sequences (e.g., tDCS → cognitive training vs. cognitive training → tDCS vs. simultaneous delivery) to elucidate the mechanisms underlying any synergistic effects.

Third, regarding neuromodulation, the advantage of using a commercial device impeded the complete implementation of the fatigue protocol with bilateral S1 stimulation as the target [42], possibly contributing to the non-significant effect of the tDCS alone. Furthermore, the lower efficacy of the tDCS in mitigating fatigue could derive from the cathode, which in our previous experience had an area twice as large as the anode, while here the two electrodes had equal dimensions, thus inducing a current density in the occipital region equal to that in the parietal region, which potentially induces inhibitory effects in visual areas.

These methodological considerations provide important guidance for future research. Specifically, we recommend the following: (1) randomized controlled trials with different intervention sequences to determine optimal treatment protocols, (2) larger-scale studies to establish population-level effects while maintaining individualized approaches, and (3) the integration of multiple outcome measures to capture the full spectrum of intervention effects. For clinical applications, our findings suggest that personalized, multimodal approaches warrant consideration, with careful attention to individual response patterns and comprehensive outcome assessments.

## 5. Conclusions

The proposed multicenter, multimodal neuromodulation and cognitive rehabilitation intervention turned out to be user-friendly, well-accepted, and with a positive responsiveness in the cognitive and motor performances of fatigued PwMS, increasing their empowerment and their quality of life.

## Figures and Tables

**Figure 1 brainsci-15-00807-f001:**
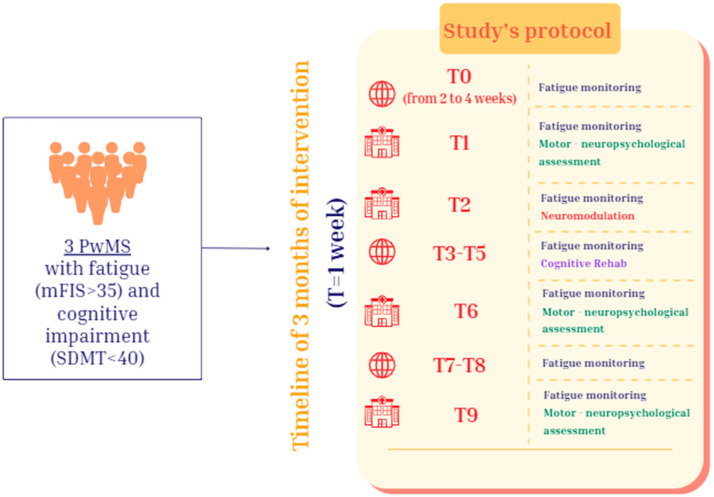
A flowchart of the protocol. The time window of each time point (T) is 1 week. The web icon is related to all the activities that the participant did alone via Google Moduli and the Rehacom rehabilitation platform, and the clinic icon is for all the activities that the participant did in the presence of the researchers. Acronyms: People with Multiple Sclerosis (PwMS), modified Fatigue Impact Scale (mFIS), Symbol Digit Modalities Test (SDMT).

**Figure 2 brainsci-15-00807-f002:**
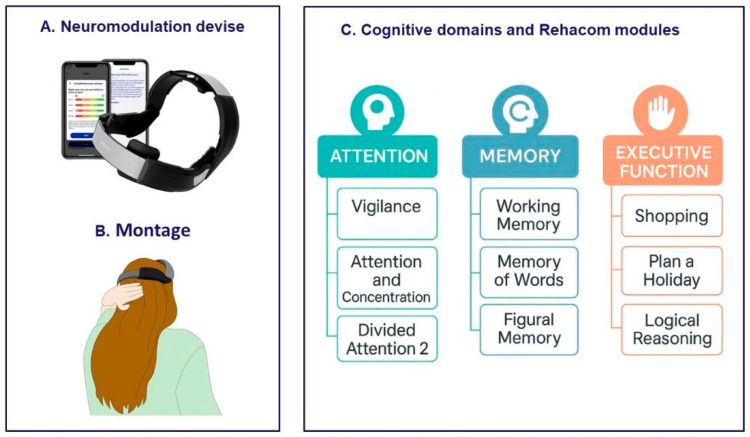
Multimodal treatment setup. (**A**) Platowork neuromodulation tDCS device and montage. (**B**) The positioning of the headset. (**C**) A graphical representation of the cognitive domains (attention, memory, and executive function) stimulated during the 3 weeks with RehaCom using, for each domain, three different devoted modules available in the software.

**Figure 3 brainsci-15-00807-f003:**
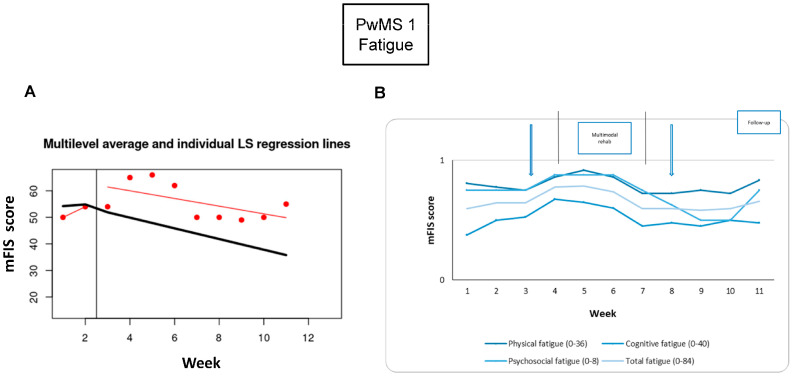
Fatigue monitoring. Each week during baseline and intervention phases: (**A**) the total fatigue levels; the Modified Fatigue Impact Scale (mFIS) collected at baseline is connected, and the vertical black line indicates the neuromodulation week; the least square linear regression (red line) is estimated after neuromodulation; (**B**) the total and the three subscales of the mFIS-quantified fatigue, normalized to the maximum subscale value [(N value/max score of the subscale), as indicated in the figure]. The arrows show the assessment times, and the vertical lines show the beginning and the end of the intervention phase. Abbreviations: People with Multiple Sclerosis (PwMS); modified Fatigue Impact Scale (mFIS).

**Figure 4 brainsci-15-00807-f004:**
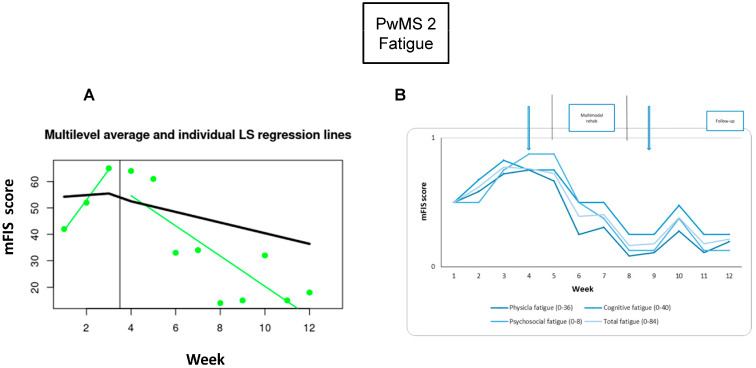
Fatigue monitoring. Each week during the baseline and intervention phases: (**A**) the total fatigue levels; the Modified Fatigue Impact Scale (mFIS) collected at baseline is connected, and the vertical black line indicates the neuromodulation week; the least square linear regression (green line) is estimated after neuromodulation; (**B**) the total and the three subscales of mFIS-quantified fatigue, normalized to the maximum subscale value [(N value/max score of the subscale), as indicated in the figure]. The arrows show the assessment times, and the vertical lines show the beginning and the end of the intervention phase. Abbreviations: People with Multiple Sclerosis (PwMS); modified Fatigue Impact Scale (mFIS).

**Figure 5 brainsci-15-00807-f005:**
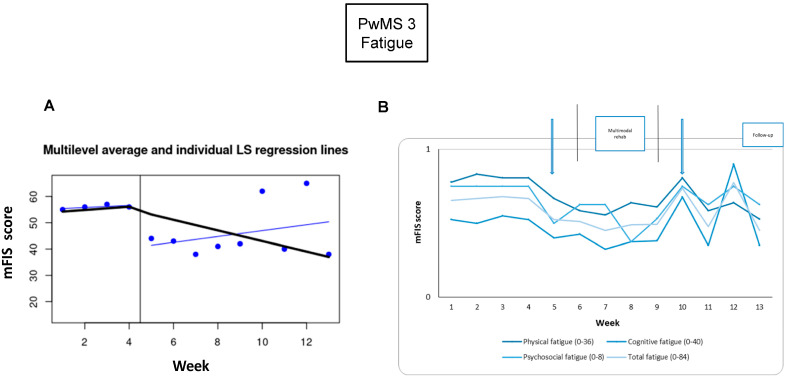
Fatigue monitoring. Each week during the baseline and intervention phases: (**A**) the total fatigue levels; the Modified Fatigue Impact Scale (mFIS) collected at baseline is connected, and the vertical black line indicates the neuromodulation week; the least square linear regression (blue line) is estimated after neuromodulation; (**B**) the total and the three subscales of mFIS-quantified fatigue, normalized to the maximum subscale value [(N value/max score of the subscale), as indicated in the figure]. The arrows show the assessment times, and the vertical lines show the beginning and the end of the intervention phase. Abbreviations: People with Multiple Sclerosis (PwMS); modified Fatigue Impact Scale (mFIS).

**Figure 6 brainsci-15-00807-f006:**
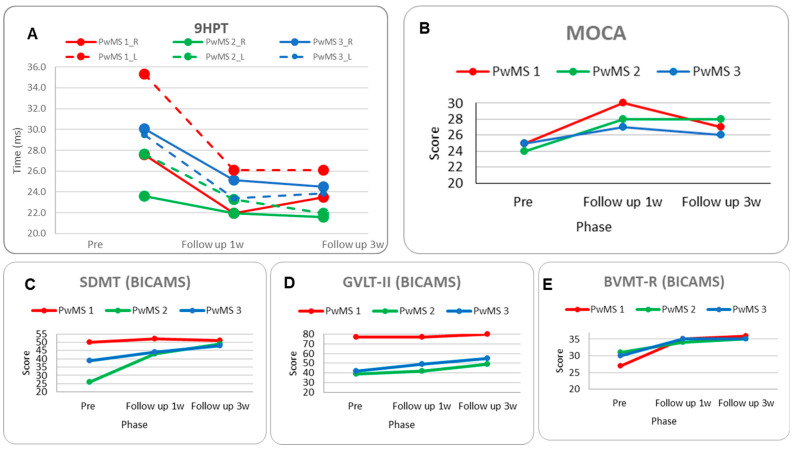
Motor and cognitive performance were assessed pre-intervention, at follow-up after 1 week, and at follow-up after 3 weeks following multimodal rehabilitation intervention. (**A**) Nine-Hole Peg Test (9HPT, ms), (**B**) Montreal Cognitive Assessment (MOCA), (**C**) Symbol Digit Modalities test (SDMT), (**D**) Greek Verbal Learning Test (GVLT-II), (**E**) Brief Visuospatial Memory Test-Revised (BVMT-R); BICAMS (Brief International Cognitive Assessment for Multiple Sclerosis).

**Table 1 brainsci-15-00807-t001:** Demographic and clinical profile of PwMS1, PwMS2, and PwMS3. Acronyms: People with Multiple Sclerosis (PwMS), Expanded Disability Status Scale (EDSS).

		PwMS1	PwMS2	PwMS3
**Demographic data**	Age	60	47	42
	Sex	F	M	M
	Education	16	14	12
	Working status	unemployed	employee	freelancer
	Social status	single	married	engaged
**Clinical data**	Years of disease	31	18	1
	EDSS	4	2.5	3
	Relapsing rate	0	0	0.67
	Disease Treatment	Glatiramer acetate	Dimethyl fumarate	Ofatumumab

**Table 2 brainsci-15-00807-t002:** Fatigue monitoring. The evaluation of differences between study phases. Note: Baseline means the average fatigue (mFIS) level before the start of the multimodal intervention (baseline and assessment pre). Negative percentage values above 20 indicate significant improvement and identify the participants as responders.

Phased Compared	PwMS 1	PwMS 2	PwMS 3
Baseline vs. Post tDCS	23	9	−20
Pre vs. Post RehaCom	−23	−77	−3
Baseline vs. Follow-up at 1 week	−5	−73	16
Baseline vs. Follow-up at 3 weeks	4	−68	−29

**Table 3 brainsci-15-00807-t003:** Motor-neuropsychological assessment. Note: All the values are raw scores. All the tests have been adapted for native Greek-speaking adults, and the demographically corrected normative data have been published—the scores represent the three rounds of assessments divided for each participant. Acronyms: Nine-Hole Peg Test (9HPT); Montreal Cognitive Assessment (MOCA); Brief International Cognitive Assessment in Multiple Sclerosis (BICAMS); Symbol Digit Modalities Test (SDMT); Greek Verbal Learning Test (GVLT-II); Brief Visuospatial Memory Test-Revised (BVMT-R); Multiple Sclerosis Quality of Life Questionnaire (MSQoL-54); Physical Composite Score (PCS); Mental Composite Score (MCS); Beck Depression Inventory (BDI).

Assesment	Test		PwMS1			PwMS2			PwMS3		
			Pre	Follow-Up 1 w	Follow-Up 3 w	Pre	Follow-Up 1 w	Follow-Up 3 w	Pre	Follow-Up 1 w	Follow-Up 3 w
**Motor** **performance**	9HPT	Dominant-hand	27.6	21.9	23.5	23.6	21.9	21.6	30.1	25.1	24.5
		Non- dominant hand	35.3	26.1	26.1	27.6	23.3	22.0	29.5	23.4	23.9
**Cognitive performance**	MOCA		25	30	27	24	28	28	25	27	26
	SDMT (BICAMS)		50	52	51	26	43	49	39	44	48
	GVLT-II (BICAMS)		77	77	80	39	42	49	42	49	55
	BVMT-R (BICAMS)		27	35	36	31	34	35	30	35	35
**Quality of life**	MSQoL-54	PCS	25.0	32.8	26.3	68.5	82.2	75.6	47.2	39.7	67.1
		MCS	22.1	33.7	35.7	31.4	66.1	51.7	48.6	30.6	78.8
		Energy subscale	0	0.48	0.48	2.4	3.84	3.84	2.88	1.92	5.28
		Cognitive subscale	9	10.5	11.25	4.50	11.25	12	11.25	9	12
**Mood**	BDI		17	15	22	23	8	19	9	11	10

## Data Availability

The data that support the findings of this study are available from the corresponding author upon reasonable request. The data are not publicly available due to privacy reasons.

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
