# Peer review of "tDCS and Cognitive Training for Fatigued and Cognitively Impaired People with Multiple Sclerosis: An SCED Study"

_brainsci, 2025, doi:10.3390/brainsci15080807_

Round 1
Reviewer 1 Report (Previous Reviewer 2)
Comments and Suggestions for Authors
This manuscript explores the combined effect of transcranial direct current stimulation (tDCS) and RehaCom software for cognitive rehabilitation in patients with fatigue and cognitive impairment due to multiple sclerosis (MS). The proposed multimodal intervention is innovative and offers a promising new direction for personalized rehabilitation in MS patients, which is commendable.
However, there are several important issues in the study:
Firstly, The sample size of the current study is too small, which may affect the reliability and stability of the conclusion.
Secondly, in the experimental design section, the combined intervention lacks cross-validation (e.g., a design where cognitive training is followed by tDCS, or separate intervention groups). This omission makes it impossible to clearly determine the independent contributions or synergistic effects of tDCS and cognitive training. Furthermore, the theoretical basis or feasibility considerations for the chosen intervention order are not discussed, which should be addressed and clarified.
Thirdly, while the RehaCom software adjusts the difficulty of tasks based on “the participant’s average performance,” the specific algorithms (such as error rate thresholds or changes in reaction times) are not provided. This makes it difficult to replicate the dose-effect relationship between training intensity and cognitive improvement. For instance, the significant improvement of PwMS2 in the memory module may be due to the scientifically incremented task difficulty, or it could simply be attributed to random fluctuations. The manuscript needs to provide more details on the algorithm to support reproducibility.
Additionally, one detail that requires attention is that fatigue was assessed solely using the mFIS, which does not differentiate between physical fatigue (e.g., the physical subscale of the Chalder Fatigue Scale) and cognitive fatigue. Given the heterogeneity of fatigue in MS patients, this could potentially lead to bias in the intervention response. This distinction should be considered and addressed.
Finally, a comprehensive revision of the manuscript’s language to ensure clarity, fluency, and consistency is strongly recommended. Professional editing will greatly enhance the manuscript’s quality and readability.
Author Response
Please see the attachment.

Reviewer 2 Report (Previous Reviewer 1)
Comments and Suggestions for Authors
The authors have addressed all the reviewers' comments satisfactorily. Therefore, I recommend that the paper be accepted for publication
Author Response
We thank the Reviewer for the appreciation of our work, sharing the relevance of the objective in mitigating invalidating symptoms and the strategic approach adopted.
Reviewer 3 Report (New Reviewer)
Comments and Suggestions for Authors
The article "Personalized tDCS and Cognitive Training for Fatigued and Cognitively Impaired People with MS: A SCED Study" explores an important issue in modern neurology. Cognitive impairment and fatigue are common symptoms of multiple sclerosis that significantly affect the quality of life of patients. The study uses a combination of two non-pharmacological methods - transcranial direct current stimulation (tDCS) and cognitive training - to address these challenges.
The main limitation of this study is small sample size (n=3). The authors chose the N=1 design, (multiple baseline single-case experimental design, mbSCED). I believe this approach is not suitable for the current context. While it may be useful for developing personalized interventions for other cases, it is not appropriate for this study. For example, the location of electrodes could be determined based on data on changes in neural networks and specific fatigue mechanisms in each patient. There have been several phenotypes identified for working memory impairment in patients with multiple sclerosis (e.g. 10.3390/jcm11102936), and these data could be used to create personalized goals for stimulation or cognitive training. However, the study did not personalize the intervention. Assessment of cognitive impairment and fatigue was challenging due to the use of subjective and unreliable indicators.
The small sample size raises concerns about the reliability and generalizability of the data presented in this article. As a result, we are unable to make definitive conclusions about the effectiveness of the methods used, the variability in outcomes among individuals, or possible causes for this variability. For these reasons, I recommend rejecting this article.
Comments on the Quality of English Languageno
Author Response
Please see the attachment.

Reviewer 4 Report (New Reviewer)
Comments and Suggestions for Authors
The study addresses an important and clinically relevant topic—the non-motor symptoms of multiple sclerosis (MS), particularly fatigue and cognitive impairment. These symptoms are often overlooked, despite having a significant impact on patients’ quality of life. The study investigates the effects of a multimodal intervention combining cognitive rehabilitation and neuromodulation to alleviate fatigue and enhance cognitive performance in individuals with MS.
The most controversial aspect of the study is the use of a Multiple Baseline Across Subjects Single-Case Experimental Design (mbSCED). While this methodology is intended to evaluate intervention efficacy at the individual level, the extremely small sample size raises concerns about the reliability and generalizability of the findings. A more robust study with a larger sample size would likely enhance the scientific quality and impact of the results and attract greater attention from the research community.
Although I understand the authors’ rationale for selecting this methodological approach, given the limited data, it is difficult to draw strong conclusions regarding the efficacy of the intervention. The authors should more carefully define the study aims in accordance with what can realistically be achieved using this design. Additionally, the methodology section is described only briefly and would benefit from a more thorough explanation and stronger justification for the chosen design.
Additional Comments:
-
Figure 1: The figure lacks clarity. Improving the contrast between the font and background, and reducing the number of colors used, would enhance readability. All abbreviations should be fully defined in figure legends, as well as in all tables and figures throughout the manuscript.
-
Medication Names: It would be preferable to use internationally recognized generic names for the medications rather than brand names.
-
Figures: Consider combining some figures or moving less critical ones to the supplementary materials. Figures should be interpretable on their own; references such as “same as in Figure 3” reduce the clarity and quality of presentation.
-
Line 455: The mention of the number of citations related to the SCED methodological approach appears unnecessary and does not add value. A more detailed description and justification of the methodological choice would be more informative and constructive.
Round 2
Reviewer 3 Report (New Reviewer)
Comments and Suggestions for Authors
I am grateful to the authors for their reply.
I actively support researches with single-case experimental designs (SCEDs), especially in the field of rehabilitation, It seems to me that the authors have made important additions to the article that will help readers better understand the specifics of the study. Unfortunately, at the moment, not many studies with this design have been conducted in MS.I hope this article will help popularize this approach.
However, I believe that my remarks are also well-founded. Although I welcome research with Single-case experimental design (SCEDs) in principle, I believe that the choice of this design in this case is quite controversial. Large-scale studies have already been conducted that have shown the efficacy of tDCS and cognitive rehabilitation methods for multiple sclerosis. There is no reason to believe that a combination of these methods could cause problems in terms of feasibility or acceptability for participants. In addition only preliminary conclusions can be drawn about the efficacy of this multimodal approach based on this study. The personalization methods used in the work, such as determining the location of the electrodes, taking into account the anatomical features of each participant, and adapting the difficulty level of tasks depending on the results of each participant, are also successfully used in large-scale randomization studies.
It is worth noting that in the current version, most of my comments have been taken into account by the authors in the restrictions.
In general, I think that the article can be published after minor revisions.
However, I think that the text of the article should provide more detailed information about the tDCS methodology (in particular, the use of individual brain MRI data to personalize the location of the electrodes and the size of the electrodes).
Author Response
Please see the attachment.

Reviewer 4 Report (New Reviewer)
Comments and Suggestions for Authors
The authors have significantly improved the justification for the methodology employed, and the study was conducted in alignment with the principles typical of SCED-type research. They have also incorporated a number of modifications based on my previous suggestions, which has strengthened the overall quality of the manuscript.
However, the methodology adopted still raises considerable concerns. In my view, it remains subject to substantial limitations. Nonetheless, within the context and constraints of this research design, the authors have adhered appropriately to the expected standards for studies of this kind.
Author Response
We would like to thank the reviewer for their assistance in improving the quality of our article through their previous suggestions, while always keeping in mind the limitations of the experimental design.
This manuscript is a resubmission of an earlier submission. The following is a list of the peer review reports and author responses from that submission.
Round 1
Reviewer 1 Report
Comments and Suggestions for Authors
I would like to thank for the opportunity to review this manuscript titled "Personalized tDCS and Cognitive Training for fatigued and cognitively impaired people with MS: A SCED Study". This study addresses a highly relevant and timely topic for people with MS. The focus on invisible symptoms, such as fatigue and cognitive impairment, is crucial, as these issues significantly impact the quality of life of people with MS and their caregivers. The manuscript is well-written and clearly structured. The figures and tables are clear, informative, and effectively support the content, facilitating understanding for the reader.
However, despite the relevance of the topic, I regret to say that I do not recommend this manuscript for publication in its current form. The primary concern relates to the extremely limited sample size (n=3), which severely undermines the reliability and generalizability of the findings. Although small-scale or pilot studies can play an important role in exploring feasibility, with such a small number of participants, it is not possible to draw any stable or meaningful conclusions regarding efficacy, nor is it feasible to explore inter-individual variability in a scientifically sound manner.
Furthermore, one area that could be further improved is the discussion of the relationship between fatigue and cognitive dysfunction in introduction paragraph. This is an intriguing and clinically relevant topic, as growing evidence suggests that these symptoms are interrelated and may share overlapping neural mechanisms. I would suggest to add further references that explore this relationship to provide a more robust conceptual framework.
I encourage the authors to consider conducting a more robust study with a larger sample size and a clearer methodological framing (e.g., feasibility or proof-of-concept), which could provide a more solid contribution to the field.
Author Response
Reviewer 1
I would like to thank for the opportunity to review this manuscript titled "Personalized tDCS and Cognitive Training for fatigued and cognitively impaired people with MS: A SCED Study". This study addresses a highly relevant and timely topic for people with MS. The focus on invisible symptoms, such as fatigue and cognitive impairment, is crucial, as these issues significantly impact the quality of life of people with MS and their caregivers. The manuscript is well-written and clearly structured. The figures and tables are clear, informative, and effectively support the content, facilitating understanding for the reader.
> We thank the Reviewer very much for the appreciation of our work, sharing the relevance of the objective in mitigating invalidating symptoms and the strategic approach adopted.
However, despite the topic's relevance, I regret to say that I do not recommend this manuscript for publication in its current form. The primary concern relates to the extremely limited sample size (n=3), which severely undermines the reliability and generalizability of the findings. Although small-scale or pilot studies can play an important role in exploring feasibility, with such a small number of participants, it is not possible to draw any stable or meaningful conclusions regarding efficacy, nor is it feasible to explore inter-individual variability in a scientifically sound manner.
I encourage the authors to consider conducting a more robust study with a larger sample size and a clearer methodological framing (e.g., feasibility or proof-of-concept), which could provide a more solid contribution to the field.
> In our international Greek–Italian multicenter team, the collaboration was particularly enriching, characterized by a fruitful exchange of complementary expertise. One group adopted the neuromodulation strategy for fatigue mitigation— tDCS for five consecutive days, 15 minutes per day, targeting parietal and occipital regions—originally proposed by the other, while the latter incorporated the study design contributed by their colleagues from the partnering country. This design, based on the Single-Case Experimental Design (SCED) framework, was introduced only recently (2018) to address the growing need for reliable evidence of efficacy at the individual rather than population level. In this context, SCED offers a strategic tool for developing personalized interventions, enabling the fine-tuning of treatments already proven effective, within populations heterogeneous in symptom expression. The relevance of this emerging methodological approach is reflected in the high impact of the foundational SCED study, which has already received over 250 citations on Scopus.
Furthermore, one area that could be further improved is discussing the relationship between fatigue and cognitive dysfunction in the introduction paragraph. This is an intriguing and clinically relevant topic, as growing evidence suggests that these symptoms are interrelated and may share overlapping neural mechanisms. I would suggest adding further references that explore this relationship to provide a more robust conceptual framework.
> Since we agree with the Reviewer that the interdependence of fatigue and cognitive dysfunction is highly relevant, we extended the related concepts in the revised Introduction section.
Reviewer 2 Report
Comments and Suggestions for Authors
The study adopted a Subjects Single-Case Experimental Design to the effect of a four-week treatment involving transcranial direct current stimulation and cognitive training using RehaCom software on Persons with Multiple Sclerosis. The results showed that the efficacy outcomes varied among participants, with two PwMS showing significant decreases in fatigue and improvements in cognitive performance. However, there are some important problems in this research. The sample size is too small, and the reliability and validity of the research results are difficult to guarantee. In addition, there are still some minor problems in writing. For example, Figure 3A lacks a legend explaining the meanings of the red and black lines; Figure 4A lacks a legend explaining the meanings of the red and green lines; Figure 5A lacks a legend explaining the meanings of the red and blue lines.
Author Response
Reviewer 2
The study adopted a single-subject single-case Experimental Design to evaluate the effect of a four-week treatment involving transcranial direct current stimulation and cognitive training using RehaCom software on Persons with Multiple Sclerosis. The results showed that the efficacy outcomes varied among participants, with two PwMS showing significant decreases in fatigue and improvements in cognitive performance. However, there are some important problems in this research. The sample size is too small, and the reliability and validity of the research results are difficult to guarantee. In addition, there are still some minor problems in writing.
> We selected the multiple baseline single-case experimental design (mbSCED), as a methodological tool previously well described and adopted in literature (Krasny-Pacini & Evans, 2018, Scopus number of citations 225). We considered this approach especially suitable for targeting the coexistence of two different symptoms, thereby personalizing intervention strategies. In particular, we moved toward the neuromodulation tDCS setting suitable for fatigue (5 days, 15 minutes per day, anodal on somatosensory areas against cathodal on occipital region), integrating cognitive rehabilitation. Notably, we adopted the statistical analysis design indicated by the Authors (Krasny-Pacini & Evans, 2018) (https://manolov.shinyapps.io software).
While we are aware of the small sample size inherent in the mbSCED approach, we emphasize that the strength of this design lies in its ability to generate valid within-person evidence. Rather than aiming for group-level statistical generalization, our goal is to contribute to advancing individualized care pathways in complex and heterogeneous clinical populations such as PwMS.
For example, Figure 3A lacks a legend explaining the meanings of the red and black lines; Figure 4A lacks a legend explaining the meanings of the red and green lines; Figure 5A lacks a legend explaining the meanings of the red and blue lines.
> We rewrite each legend to explain the meaning of each line for all participants.
Reviewer 3 Report
Comments and Suggestions for Authors
The manuscript presents a highly relevant and well-structured investigation into the combined effects of transcranial direct current stimulation (tDCS) and cognitive rehabilitation using RehaCom software in individuals with Multiple Sclerosis (MS) suffering from fatigue and cognitive impairments. The adoption of a multiple baseline single-case experimental design (mbSCED) is particularly commendable for its suitability in individualized interventions and for addressing the heterogeneity inherent to MS populations.
The study is conceptually robust, clinically grounded, and methodologically detailed. The integration of motor, cognitive, emotional, and experiential outcomes reflects a comprehensive biopsychosocial approach to MS care. Nonetheless, the manuscript would benefit from a professional language revision. There are recurring grammatical inconsistencies, awkward phrasings, and incorrect uses of syntax and articles that, although not severely compromising comprehension, detract from the overall academic quality. Examples include improper subject-verb agreement, long and convoluted sentences, and inconsistent tense usage, particularly in the methods and results sections.
In addition, the manuscript exhibits traces of literal translation from other languages into English, which results in unusual constructions such as “amelioration of fatigue” or “fatiguing experience for finalizing the session.” Phrases like “the participant who benefitted overall had two moments of fatigue worsening” would be clearer as “the participant who benefited overall experienced two episodes of increased fatigue.”
Beyond language, the paper would also benefit from refinement in its scientific writing style. There is a tendency toward verbose descriptions, and certain interpretations in the discussion could be streamlined or more cautiously worded. Moreover, while the rationale for the sequential introduction of interventions is clear, the effects of tDCS and cognitive training remain somewhat confounded.
A clearer delineation of their respective contributions, even through speculative discussion or correlation analysis, would enhance interpretability. The explanation of how Minimal Clinically Important Differences (MCIDs) were operationalized for each outcome is appreciated, though it would be helpful to clarify whether these thresholds were based on literature or study-specific criteria.
In terms of formatting, some figure and table legends could be made more concise and standardized. Additionally, abbreviations should be defined upon first use in both abstract and main text, and some (e.g., PwMS) appear before being explained.
In summary, this is a promising and methodologically sound contribution to the literature on non-pharmacological interventions in MS. With a thorough language edit and modest refinement of the discussion and presentation, the manuscript will be suitable for publication.
Comments on the Quality of English LanguageThe manuscript presents meaningful and clinically relevant findings; however, a comprehensive English language revision is necessary to ensure clarity, fluency, and consistency throughout the text. Several sections display grammatical errors, awkward or literal phrasing—likely resulting from translation—and inconsistent use of verb tenses. For instance, expressions such as “amelioration of fatigue,” “fatiguing experience for finalizing the session,” and “the participant who benefitted overall had two moments of fatigue worsening” should be revised to more natural and academically appropriate constructions like “reduction in fatigue,” “fatiguing experience when completing the session,” and “the participant who benefited overall experienced two instances of increased fatigue,” respectively. The manuscript also contains overly long and complex sentences that could be split for better readability. Articles and prepositions are occasionally misused, and some words are repeated unnecessarily. Additionally, consistent use of terminology and abbreviations—such as defining PwMS at first mention—is recommended. A professional copyedit will substantially enhance the quality and readability of the manuscript, aligning it with the expectations of an international academic audience.
Author Response
Reviewer 3
The manuscript presents a highly relevant and well-structured investigation into the combined effects of transcranial direct current stimulation (tDCS) and cognitive rehabilitation using RehaCom software in individuals with Multiple Sclerosis (MS) suffering from fatigue and cognitive impairments. The adoption of a multiple baseline single-case experimental design (mbSCED) is particularly commendable for its suitability in individualized interventions and for addressing the heterogeneity inherent to MS populations.
> We sincerely thank the Reviewer for appreciating the robustness and relevance of our study design. We are particularly glad that the value of the multiple baseline single-case experimental design (mbSCED) has been recognized. This approach is especially suitable for addressing the heterogeneity of people with Multiple Sclerosis (PwMS), and for targeting the coexistence of two impactful symptoms—chronic fatigue and cognitive impairment—through personalized intervention strategies.
Indeed, the rationale of our work is not to demonstrate the superiority or general efficacy of one specific treatment, but rather to provide a rigorous methodological framework capable of identifying therapeutic tools suitable for a given individual. The mbSCED allows us to test established interventions—each addressing motor, cognitive, or emotional dimensions—in a personalized, integrative manner consistent with a biopsychosocial model of care.
The study is conceptually robust, clinically grounded, and methodologically detailed. Integrating motor, cognitive, emotional, and experiential outcomes reflects a comprehensive biopsychosocial approach to MS care. Nonetheless, the manuscript would benefit from a professional language revision. There are recurring grammatical inconsistencies, awkward phrasings, and incorrect uses of syntax and articles that, although not severely compromising comprehension, detract from the overall academic quality. Examples include improper subject-verb agreement, long and convoluted sentences, and inconsistent tense usage, particularly in the methods and results sections.
In addition, the manuscript exhibits traces of literal translation from other languages into English, which results in unusual constructions such as “amelioration of fatigue” or “fatiguing experience for finalizing the session.” Phrases like “the participant who benefited overall had two moments of fatigue worsening” would be clearer as “the participant who benefited overall experienced two episodes of increased fatigue.”
> We also value the Reviewer’s suggestions regarding the language of the manuscript. We have carefully revised the text to improve fluency, clarity, and grammatical consistency, especially in the methods and results sections. We believe these changes enhance the overall academic quality of the manuscript, without altering its scientific content.
Beyond language, the paper would also benefit from refinement in its scientific writing style. There is a tendency toward verbose descriptions, and certain interpretations in the discussion could be streamlined or more cautiously worded. Moreover, while the rationale for the sequential introduction of interventions is clear, the effects of tDCS and cognitive training remain somewhat confounded.
> We refined the discussion, making more consisten the results obtained with the study design adopted.
We acknowledge the Reviewer's concern regarding the potential confounding between the effects of tDCS and cognitive training. In line with the rationale of the SCED framework, our study aims to explore the benefit of addressing a dual source of suffering—fatigue and cognitive impairment—that are often interdependent in PwMS. While we are aware that isolating the specific contributions of each intervention would require a different, dedicated experimental design, we opted for a sequential introduction guided by clinical reasoning.
Specifically, we prioritized the intervention on fatigue, based on the understanding that fatigue represents a global state affecting physical, cognitive, and social functioning. By first targeting this overarching condition through neuromodulation, we anticipated that a general increase in available energy would enhance the individual’s capacity to engage meaningfully in the subsequent cognitive rehabilitation phase. This rationale aligns with a personalized, biopsychosocial approach, which we consider essential in complex clinical populations.
A clearer delineation of their respective contributions, even through speculative discussion or correlation analysis, would enhance interpretability. The explanation of how Minimal Clinically Important Differences (MCIDs) were operationalized for each outcome is appreciated, though it would be helpful to clarify whether these thresholds were based on literature or study-specific criteria.
>We revised the thresholds and provided the reference for the clinical significance thresholds for the different dimensions modified by the interventions.
In terms of formatting, some figure and table legends could be made more concise and standardized. Additionally, abbreviations should be defined upon first use in both the abstract and the main text, and some (e.g., PwMS) appear before being explained.
> We rewrite the legend of the figures and define all the abbreviations at their first use.
In summary, this is a promising and methodologically sound contribution to the literature on non-pharmacological interventions in MS. With a thorough language edit and modest refinement of the discussion and presentation, the manuscript will be suitable for publication.
Comments on the Quality of English Language
The manuscript presents meaningful and clinically relevant findings; however, a comprehensive English language revision is necessary to ensure clarity, fluency, and consistency throughout the text. Several sections display grammatical errors, awkward or literal phrasing—likely resulting from translation—and inconsistent use of verb tenses. For instance, expressions such as “amelioration of fatigue,” “fatiguing experience for finalizing the session,” and “the participant who benefitted overall had two moments of fatigue worsening” should be revised to more natural and academically appropriate constructions like “reduction in fatigue,” “fatiguing experience when completing the session,” and “the participant who benefited overall experienced two instances of increased fatigue,” respectively. The manuscript also contains overly long and complex sentences that could be split for better readability. Articles and prepositions are occasionally misused, and some words are repeated unnecessarily. Additionally, consistent use of terminology and abbreviations—such as defining PwMS at first mention—is recommended. A professional copyedit will substantially enhance the quality and readability of the manuscript, aligning it with the expectations of an international academic audience.
>We sincerely appreciate your thorough review and your helpful comments regarding the language and lexical aspects of the manuscript.
In response to your observations, we have undertaken a comprehensive revision of the English language and terminology used throughout the text. Grammatical errors, awkward phrasing, and inconsistent verb tenses have been corrected to ensure clarity, fluency, and coherence. Additionally, we have simplified overly long sentences to improve readability, corrected the use of articles and prepositions, reduced unnecessary repetition, and ensured consistent terminology and abbreviation usage.
We are confident that these revisions have significantly improved the manuscript's readability and alignment with international academic standards.